# Potassium Bioaccessibility in Uncooked and Cooked Plant Foods: Results from a Static In Vitro Digestion Methodology

**DOI:** 10.3390/toxins14100668

**Published:** 2022-09-26

**Authors:** Costanza Ceccanti, Lucia Guidi, Claudia D’Alessandro, Adamasco Cupisti

**Affiliations:** 1Department of Agriculture, Food and Environment, University of Pisa, 56124 Pisa, Italy; 2Interdepartmental Research Center Nutrafood “Nutraceuticals and Food for Health”, University of Pisa, 56124 Pisa, Italy; 3Department of Clinical and Experimental Medicine, University of Pisa, 56126 Pisa, Italy

**Keywords:** plant-based diet, in vitro food digestion, potassium bioaccessibility, food boiling, CKD, ESKD, potassium intake, hyperkalemia

## Abstract

Hyperkalemia is a major concern in chronic kidney disease and in end-stage renal disease, representing a predictor of hospitalization and mortality. To prevent and treat hyperkalemia, dietary management is of great clinical interest. Currently, the growing use of plant-based diets causes an increasing concern about potassium load in renal patients. The aim of this study was to assess the bioaccessibility of potassium in vegetables, concerning all aspects of the plants (fruit, flower, root, tuber, leaf and seed) and to what extent different boiling techniques affect potassium content and bioaccessibility of plant-based foods. Bioaccessibility was evaluated by an in vitro digestion methodology, resembling human gastro-intestinal tract. Potassium content was higher in seeds and leaves, despite it not being possible to define a common “rule” according to the type of organ, namely seed, leaf or fruit. Boiling reduced potassium content in all vegetables excluding carrot, zucchini, and cauliflower; boiling starting from cold water contributed to a greater reduction of the potassium content in potato, peas, and beans. Bioaccessibility after in vitro digestion ranged from 12 (peas) to 93% (tomato) regardless of species and organs. Higher bioaccessibility was found in spinach, chicory, zucchini, tomato, kiwi, and cauliflower, and lower bioaccessibility in peas. Potassium from leaf resulted in the highest bioaccessibility after digestion; as a whole potassium bioaccessibility in the fruits and vegetables studied was 67% on average, with differences in relation to the different organs and species. Further, considering the method of boiling to reduce potassium content, these data indicate that the effective potassium load from plant-based foods may be lower than originally expected. This supports the clinical advices to maintain a wide use of plant-based food in the management of renal patients.

## 1. Introduction

Hyperkalemia is of great concern in chronic kidney disease (CKD) and end-stage-kidney-disease (ESKD) patients, due to the risk for life-threating arrhythmias [1]. Several factors challenge the preservation of potassium (K) homeostasis in CKD, leading to high risk of hyperkalemia [2]. The progressive decline of kidney function leads to impaired capacity of K excretion, favoring the risk of hyperkalemia, as well as the presence of diabetes mellitus, the use of renin-angiotensin system blockers and metabolic acidosis [3]. To counteract hyperkalemia, dietary K restriction is recommended, usually reducing the intake of fruits and vegetables, due to their high content of dietary K [4,5]. However, K is ubiquitous and also present in animal-based foods as an intracellular component. It is notable that plant cell differs from animal cell due to the presence of the cell-walls. The difficult digestion of plant cell walls by humans and the K intracellular localization may affect the K bioaccessibility and bioavailability that could result lower in plant food compared to animal food [6]. Bioaccessibility is the amount of an ingested nutrient that is available for absorption in the gut after digestion, whilst bioavailability is the certain quantity of a nutrient that reaches the site of physiological activity [7]. The cell-wall of plant foods creates a physical barrier that decreases the bioaccessibility of intracellular nutrients as K [6]. For these reasons, dietary restriction of fruits and vegetables could not be effective in reducing the effective K load and may paradoxically increase the risk of hyperkalemia for the concomitant restriction of fibers and alkali content [8].

Plant-based food is represented by vegetables, fruits, cereals, legumes, seeds, and nuts and, among them, human utilizes several plant organs: stem (i.e., celery or asparagus), leaves (i.e., spinach, lettuce, chicory), inflorescence (i.e., cauliflower), petiole (i.e., celery branches), tubers (i.e., potato), roots (i.e., carrot), fruits (i.e., tomato, apple, banana) and seeds (i.e., wheat, pea) [9]. Plant-based food contains macronutrients (carbohydrate, proteins and lipids), micronutrients (vitamins and minerals) but also dietary fibers and phytochemicals (i.e., phenols, phytosterols) [10]. The percentages of nutrients change within plant organs due to their different roles in the plant development and growth. Evidently, plant organs are constituted by different tissues in which there are various types of cell, and each cell is delimited by a cell wall which joins it to its surrounding cells. Once harvested, the diversity remains, and it has a major impact on the bioaccessibility of nutrients. The plant cell wall is crucial for nutrient bioaccessibility. Indeed, the cell wall is composed of cellulose microfibrils integrated within a gel-type matrix typically comprising pectin, hemicellulose, and small amounts of glycoproteins, phenolic acids, minerals and, in some cells, lignin [11]. The degree of pectin esterification determines the biochemical properties and porosity of polysaccharides within the cell wall, that, in turn, influences the diffusion of digestive enzymes and bile salts inside the cell, thus, affecting macronutrient digestion, due to digestive enzymes such as amylase, proteases and pancreatic lipase, which cannot diffuse through the cell wall in various plant-based food (wheat, chickpea, pea, mung bean, red kidney bean) while some cell walls are more permeable (common bean, potato, banana) [12]. Li et al. [12] reported that, in addition to plant species, porosity of cell wall varies along the different organs (i.e., legumes have less porous cell walls than potatoes) [12]. All these characteristics influence the physical and biochemical process of digestion [13]. The rupture of the cell wall is an important requisite for the digestion, causing the release of nutrients into the extracellular environment and the diffusion of the enzymes through a permeable cell wall. However, the cell wall matrix and individual cell wall polysaccharides in plants behave in a similar way during digestion. The cell walls characterized by a high permeability or more prone to physical disruption (for example during chewing) release macro- and micro-nutrients easily, rendering them more bioaccessible [14]. Thus, despite the high content of some macro- or micro-nutrients as well as minerals of some plant-based food, their accessibility to digestion within the human gastro-intestinal tract varies in relation to the plant species and the organ used as food.

Since hyperkalemia is a predictor and also a cause of hospitalization and mortality especially in CKD or ESKD population, hyperkalemia prevention and treatment by dietary management is of great clinical interest. Plant foods play important roles in CKD diet for their alkali load and fiber content [15]. Major knowledge about K bioaccessibility of plant food may help in modifying diet for CKD patients, without an exclusion of plant-based food intake. 

The aim of the present study was to assess the bioaccessibility of K from some fruits and vegetables when utilized as plant-based food. In addition, we explored the effects of different boiling techniques on K content and bioaccessibility of the same plant foods.

## 2. Results

### 2.1. Potassium Content in Raw Samples

Figure 1 shows the different K content in the fresh samples of different species and organs analyzed in this study. Peas had the highest value, whereas carrot, as a root, contained the lowest level of K. Among fruits, the highest K concentration was found in avocado. Seed and leaf had higher K content than the other organs under investigation. 

Species under investigation differ in the edible part, but it is not possible to find a common behavior in relation to the plant organ utilized as food. Indeed, completely different organs such as potato, zucchini and tomato have similar K content.

### 2.2. Effect of Boiling Procedures

In Table 1 content of K after boiling procedures for fruits, vegetables and legumes is reported. It is evident for all samples that K content decrease when compared to the raw samples as reported in Figure 1. It is suggested that the reduction in K content is represented by the leached K in water (not determined in this report).

As a whole, K content decreased after boiling but differences were found in relation to the applied boiling procedures for exception of peas, spinach, cauliflower, zucchini, beans, and carrot (Figure 2). For carrots, boiling starting from hot water induced a higher reduction of K content compared to starting from cold water.

### 2.3. Potassium Bioaccessibility of Raw Samples

Figure 3 reports the bioaccessible K percentage after the in vitro digestion. K bioaccessibility ranged from 12 (pea) to 93% (tomato) in different species/organs. The highest bioaccessibility of K was found in spinach, zucchini, and tomato, followed by kiwifruit. The lowest percentage of bioaccessible K was found in peas, although this seed had the highest content in K in raw material. There is no evident relationship between K bioaccessibility and its organ. As a whole, the average value of K bioaccessibility in fruits and vegetables is close to 67%, specifically 63.5% in fruits, and 71.7% in the analyzed vegetables.

### 2.4. Effect of Boiling on Potassium Bioaccessibility

Table 2 demonstrates nutritional guideline in regard to K content expressed as mg on 100 g FW in digested samples. It is well-demonstrated that boiling affected K bioaccessibility in the studied samples (without K content quantification in boiled water). However, it was not possible to identify a characteristic pattern for each plant organ (Figure 4).

Peas had the highest K concentration in raw samples, the lowest bioaccessibility as raw product but its bioaccessibility widely increased after boiling, and in particular starting from hot water (Figure 4). On the contrary, a low K content was recorded in carrots even though a high bioaccessibility was found as raw products. When boiled starting from cold water, the K bioaccessibility of carrot increased whilst decreased starting from hot water.

## 3. Discussion

Dietary K manipulation is a major concern in renal nutrition practice. The risk of hyperkalemia is well known in CKD and ESKD patients. Thus, the restriction of K intake is often prescribed as reducing intake of fruit and vegetables which, on opposite, may protect form hyperkalemia by their alkali and fiber load [8,15]. Recently, the interest in the bioavailability of dietary potassium has been re-discovered even though the results are not yet satisfying. Some authors carried out in vivo studies using low and high K intake and measuring K urine excretion [16,17]. Urinary K is being considered as a measurement that generally underestimate dietary K intake [18,19,20]. For example, McDonald-Clarke et al. [16], observed high to similar bioavailability of K from white unfried potatoes as from fried French potatoes supplemented with K-gluconate, showing the greater K bioavailability from potatoes compared with that of K-gluconate. The same authors showed that frying did not change K bioavailability in processed food. In addition, Naismith and Braschi [17] carried out a 10-day trial providing unprocessed fruits and vegetables (low bioaccessible diet) or a potentially high bioaccessible diet based on animal food or on processed fruits. Analyzing the urinary K excretion, 96.3% was recovered in urine of patients with the high bioaccessible diet against 76.8% in those on the low bioaccessible diet. These authors hypothesized that these differences could be due to the different cellular structure of plant food, confirming the fundamental role of plant cell wall. The authors concluded that the bioaccessibility of K in unprocessed plant foods resulted lower than that of processed plant food, likely due to the disruption of cell wall by heat process in contrast to results obtained by McDonald-Clarke et al. [16].

In the present study, the bioaccessibility of K resulted dependent to the plant matrix but independent to the plant organ. Leaves resulted the plant structures more susceptible to the release of K during the digestion process, likely due to the low pH in the gastric phase and the dependence of Na^+^/K^+^ pump to pH [21]. Results of K bioaccessibility were in agreement to a previous study [22], which reported that a diet containing minimally processed fresh fruits and vegetables had a K bioavailability of 77%, confirming our result which indicates that K bioaccessibility in investigated fruits and vegetables was around 65–70%. Conversely to the plant matrix, no correlation between K bioaccessibility and plant organ was found. As an example, both peas and beans, belongs to legumes and are both seeds. However, they did not show any similarity in bioaccessible K. Nevertheless, further studies are needed to understand in depth their specific patterns.

An important factor that influences the bioaccessibility and bioavailability of K is the presence of other nutrients and constituents in food [23]. Among others, protein, amino acids, lipids and carbohydrates can affect the bioaccessibility of minerals forming soluble and insoluble complexes with minerals within the gastrointestinal tract [24]. It has been reported that formation of these complexes is facilitated by the presence of phytic acid, oxalates, and tannins, but inhibited by lactic acid, citric acid, fructose and ascorbic acid [24,25]. Despite these efforts, the behavior of these organic molecules on the bioaccessibility of minerals must be still elucidated. 

It is believed that thermal treatment increases the bioaccessibility of minerals due to softening release of protein-bound minerals of the plant food matrix [26,27] and alteration of solubility inhibitors such as oxalates, phytates, tannins and phenols [28]. On the other hand, the decrease of K content due to boiling processes can be explained by the disruption of cell walls and release of K in the boiling water [17]. However, this behavior was strictly dependent on the plant matrix of vegetables. Additionally, our findings in regards to a higher efficiency in reducing K in plant products by boiling is in agreement with results of Tsaltas [29]. Analyzing potatoes, this author reported that K reduction for sliced potatoes after boiling was approximately 50%, whilst reduction with leaching and boiling was approximately 80% [29]. Only the leaching process cannot be recommended in fruits or vegetables that are consumed as fresh, since Bethke and Jansky [30] did not find significant differences in K content in potatoes left in water overnight compared to the control.

## 4. Conclusions

Potassium content is higher in seeds and leaves, despite it not being possible to define a common “rule” according to the type of organ (seed, leaf or fruit). Different organs such as potato (tuber), zucchini and tomato (fruit) have similar K content. Bioaccessibility after in vitro digestion ranges from 12 to 93% regardless of species and organ. The highest bioaccessibility was found in spinach, zucchini and tomato, and the lowest bioaccessibility in peas (which instead have the highest K content in the fresh product). K from leafy samples were the most accessible after digestion. Boiling starting from cold water, instead of hot water, contributed to a greater reduction of K content in chicory, kiwifruit, potatoes and avocado, while there was no difference between the two treatments for peas, spinach, cauliflower, zucchini and beans.

In summary, these data suggest that K bioaccessibility in plant-based food is not as high as reported by previous works, and it varies among the different studied plant-based food, being roughly 65–70% in fruits and vegetables. Further, considering the potassium removal by boiling, these data indicate that the effective potassium load from plant-based foods may be lower than recently supposed. This supports the clinical advice to maintain a wide use of plant-based food in the management of renal patients.

## 5. Materials and Methods

### 5.1. Plant Material

Common fruits (banana, kiwifruit, avocado) vegetables (carrot, potato, spinach, tomato, cauliflower, chicory and zucchini) and legumes (beans and peas) were purchased in a local supermarket in Pisa (Italy). An aliquot of plant material was dried in a ventilated oven (Memmert GmbH Co. KG Universal Oven UN30, Schwabach, Germany) at 105 °C until it reached constant weight for the determination of the dry weight (DW). Another aliquot was frozen in liquid nitrogen and stored at −80 °C until the performance of in vitro digestion methodology. The last aliquot of plant material was immediately prepared for the boiling process and then subjected to the in vitro digestion methodology.

### 5.2. Boiling Procedures

The boiling process followed two procedures. In the first, an aliquot (65 g) of plant sample was placed in 500 mL of boiling milli-Q H_2_O (hot water) for different times based on edibility and palatability of fruit/vegetable/legume and plant organ under investigation: 5 min for banana, kiwi, avocado, spinach and tomato, 10 min for chicory and zucchini and 20 min for carrot, potato, cauliflower, beans and peas. It was the minimum cooking time to reach softness, palatability, and taste according to the Italian consumption habits. For each fruit/vegetable/legume, the best boiling conditions were determined in a preliminary experiment, in which the palatability was evaluated by trained panelists (50% men and 50% women) through a hedonic test. In the second procedure, a portion (65 g) of plant sample was placed in 500 mL of cold milli-Q H_2_O (cold water) and the same cooking times of the first boiling methodology were used starting from the boiling point of the water containing plant sample. After both boiling treatments, the boiling water was drained off for 60 s. An aliquot of boiled material was used to determine the retention or leaching of water during the heat treatment, performing the same procedure utilized for the DW determination. Boiled samples from both boiling treatments were frozen in liquid nitrogen and stored at −80 °C until the performance of in vitro digestion protocol.

### 5.3. In Vitro Digestion Protocol

The determination of the K bioaccessibility in plant samples was carried out following the INFOGEST in vitro digestion method published by Minekus et al. [31]. An amount (2 g) of fresh or boiled samples were subjected to simulated gastric fluid composed by 6.9 mM KCl, 0.9 mM KH_2_PO_4_, 25 mM NaHCO_3_, 47.2 mM NaCl, 0.1 mM MgCl_2_(H_2_O)_6_, 0.5 mM (NH_4_)_2_CO_3_. Pepsin and CaCl_2_ were added to the mixture to achieve a final concentration of 2000 U mL^−1^ and 0.075 mM respectively. Hydrochloric acid (HCl, 6 M) was then used to acidify the mixture to pH 3 and milli-Q H_2_O was added to reach a final volume of 20 mL. Samples were then incubated in a stirring water bath at 37 °C and 95 rpm for 2 h. Then, the pH was checked and adjusted if necessary. The simulated intestinal fluid composed by 6.8 mM KCl, 0.8 mM KH_2_PO_4_, 85 mM NaHCO_3_, 38.4 mM NaCl, 0.33 mM MgCl_2_(H_2_O)_6_ was then added together with pancreatin (100 U mL^−1^) and bile salts for a final concentration of 10 mM. CaCl_2_ was also added to achieve a final concentration of 0.3 mM. NaOH 1 M was used to bring the pH to 7 and the necessary amount of water added to reach a final volume of 20 mL. Samples were incubated in a stirring water bath at 37 °C and 95 rpm for 2 h. At the end of the incubation, aliquots (5 mL in triplicate) of each sample were centrifuged at 14,000× *g* using a laboratory centrifuge (MPW-260R, MWP Med. Instruments, Warsaw, Poland) for 20 min and the supernatant was stored at 4 °C until the K determination.

### 5.4. K Determination

K determination was performed on mineralized samples with nitric acid 70% (*v*/*v*) and H_2_O_2_ 30% (*v*/*v*), using an atomic absorption spectrometer Thermo Scientific ICE 3000. Results of raw and boiled (digested and undigested) samples were expressed as mg K per 100 g fresh weight (FW) and results of raw digested samples were expressed as percentage of bioaccessible K (calculated on mg K on volume of digestate). K content found in boiled undigested and digested samples were expressed as percentage variation compared with raw samples.

### 5.5. Statistical Analysis

All the data have been processed using the statistical software GraphPad (GraphPad, La Jolla, CA, USA). Results are reported as mean ± standard deviation of three replicates. Data of raw undigested and digested samples were analyzed by one-way ANOVA using the different species as variability factors. Means were separated by Fisher’s least significant difference (LSD) *post-hoc* test (*p* < 0.05). Data were tested for normal distribution by Shapiro–Wilk test and the variance homogeneity (homoscedasticity) was verified with Bartlett’s test. For the comparison between the different boiling procedures Student’s *t* test was applied (*p* = 0.05).

## Figures and Tables

**Figure 1 toxins-14-00668-f001:**
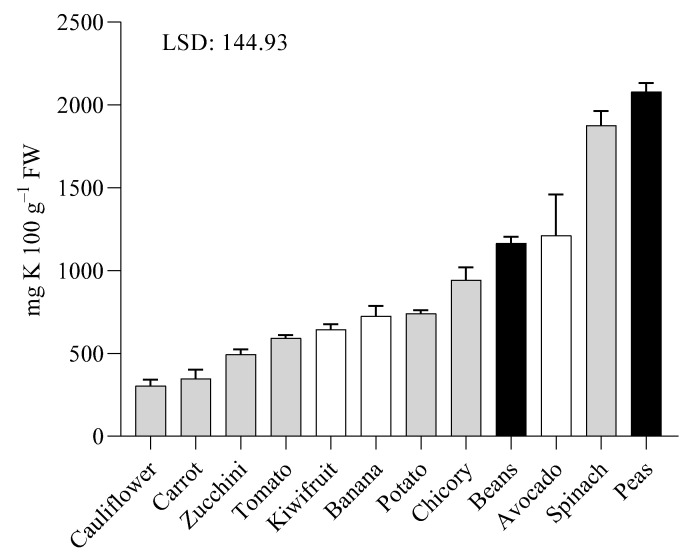
Potassium content in 100 g of raw samples of fruits (white columns), vegetables (grey columns) and legumes (black columns). Data were statistically analyzed following one-way ANOVA using the species as the variability factors and separated by Fisher’s least significant difference (LSD) *post-hoc* test (*p* < 0.05).

**Figure 2 toxins-14-00668-f002:**
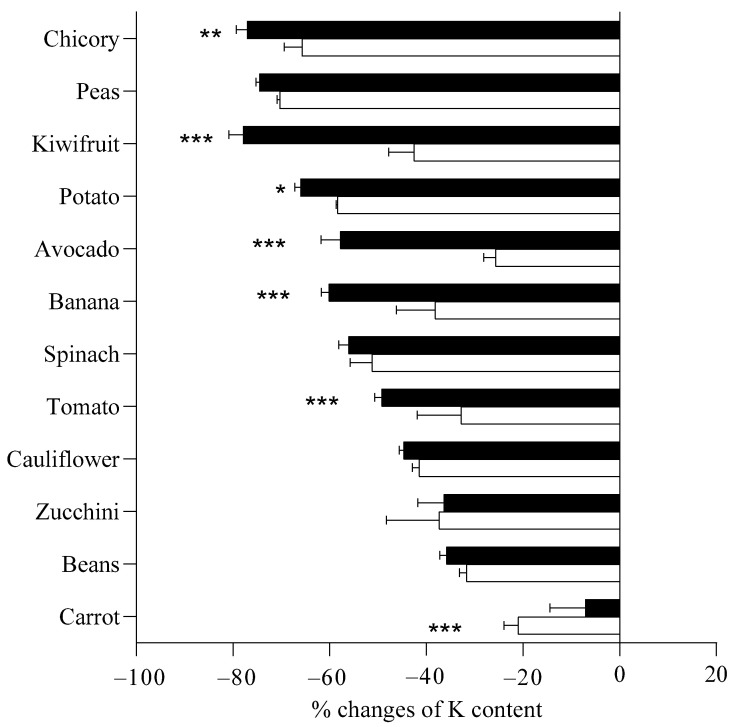
Percentage reduction of potassium content in samples when boiled starting from cold water (full columns) and when boiled starting from hot water (empty columns), in respect to potassium content in raw samples. Considering each sample singularly, means were statistically analyzed by Student’s *t* test, comparing different boiling procedures. * *p <* 0.05; ** *p <* 0.01; *** *p* < 0.001.

**Figure 3 toxins-14-00668-f003:**
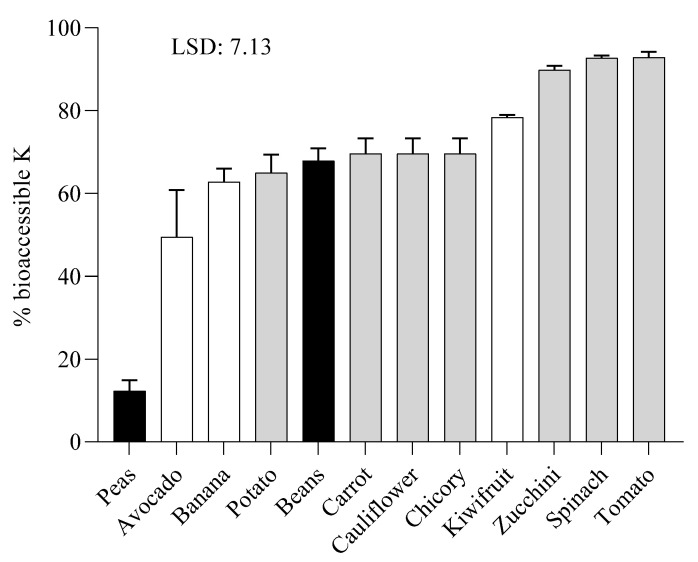
Percentage (%) of bioaccessible potassium in digested raw samples of fruits (white columns), vegetables (grey columns) and legumes (black columns). Means were statistically analyzed following one-way ANOVA using the species as the variability factors and separated by Fisher’s least significant difference (LSD) post hoc test (*p* < 0.05).

**Figure 4 toxins-14-00668-f004:**
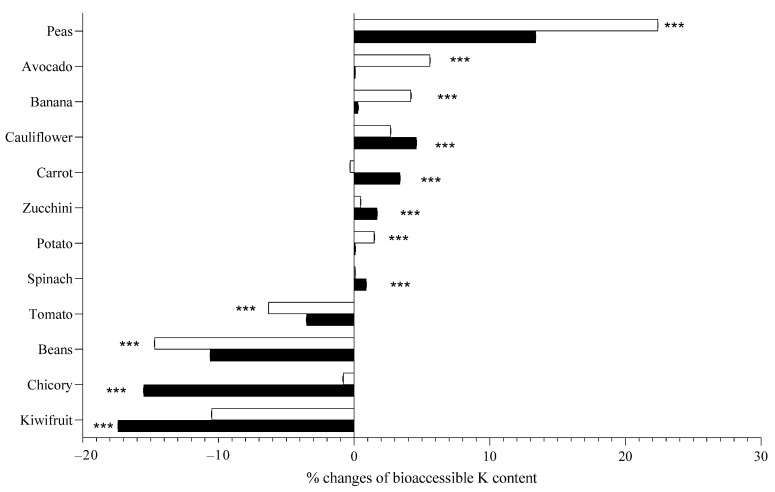
Percentage variation of bioaccessible potassium in samples analyzed when boiled starting from cold water (full columns) and when boiled starting from hot water (empty columns), when compared with potassium content in raw samples. Considering each sample singularly, means were statistically analyzed by Student’s *t* test, comparing different boiling procedures. *** *p* < 0.001.

**Table 1 toxins-14-00668-t001:** Potassium content (mg) in 100 g of boiled samples of fruits, vegetables and legumes.

	Boiling Procedures	
	Cold Water	Hot Water
Chicory	215.8	322.7
Peas	529.5	616.3
Kiwifruit	142.1	369.8
Potato	251.5	308.3
Avocado	510.9	899.9
Banana	288.9	447.8
Spinach	823.2	914.5
Tomato	300.9	398.4
Cauliflower	168.8	178.5
Zucchini	314.3	309.4
Beans	746.9	794.9
Carrot	374.9	311.3

**Table 2 toxins-14-00668-t002:** Bioaccessible Potassium content (mg) in 100 g of digested samples of fruits, vegetables and legumes.

	Raw	Cold Water	Hot Water
Peas	153.7	46.5	41.2
Avocado	91.2	39.5	65.7
Banana	40.2	32.7	38.2
Cauliflower	49.8	18.8	30.0
Carrot	52.5	32.2	47.2
Zucchini	54.3	16.9	29.3
Potato	66.1	8.7	13.6
Spinach	117.4	39.4	48.7
Tomato	30.4	29.1	95.8
Beans	31.5	26.9	46.4
Chicory	95.3	54.7	28.3
Kiwifruit	39.9	12.8	33.8

## Data Availability

Not applicable.

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
