# Peer review of "Potassium Bioaccessibility in Uncooked and Cooked Plant Foods: Results from a Static In Vitro Digestion Methodology"

_toxins, 2022, doi:10.3390/toxins14100668_

Round 1

Reviewer 1 Report

The authors investigated potassium bioaccessibilty from 12 different raw and cooked plants. Bioaccessibilty was tested by exposing plants to an artificial gastric milieu. The average bioaccessibilty lay around 70% but differed greatly between plants. Peas had the highest potassium content, but the lowest bioaccessibilty in raw form.

Major

1.       Figure 4: When the authors state that bioaccessibilty in boiled peas increased by 15 (boiled starting from cold) to 22% (boiling starting from hot water), did they measure potassium only in the cooked plant or also in the cooking water?

2.       In Figure 2, it is stated that cooking reduced the potassium content of peas by 70%. What is the absolute amount that will be taken up after ingesting 100 g of cooked peas (discarding the water?)
Is it 2000 mg (absolute potassium content; Figure 1)
minus  70% which is lost in the cooking water (Figure 2)
minus 88% (Figure 3 bioaccessibilty raw food 12%) + 22%  (Figure 4 variation of accessibility after boiling). 12% bioaccessibilty raw + 22% cooked results in 15%)
This would mean cooked peas deliver only about 90 mg potassium per 100 g.
This information could be added for all food in an extra table 5 because it is of general interest. Every nutrition guideline I know of marks peas as undesirable in hemodialysis patients

3.       The manuscript could have been proofread by a native English speaker

Author Response

Reviewer no. 1

The authors investigated potassium bioaccessibilty from 12 different raw and cooked plants. Bioaccessibilty was tested by exposing plants to an artificial gastric milieu. The average bioaccessibilty lay around 70% but differed greatly between plants. Peas had the highest potassium content, but the lowest bioaccessibilty in raw form.

Major

1.Figure 4: When the authors state that bioaccessibilty in boiled peas increased by 15 (boiled starting from cold) to 22% (boiling starting from hot water), did they measure potassium only in the cooked plant or also in the cooking water?

A... No, we have not determined K content in boiled water as we have specified in the text.

  1. In Figure 2, it is stated that cooking reduced the potassium content of peas by 70%. What is the absolute amount that will be taken up after ingesting 100 g of cooked peas (discarding the water?)
    Is it 2000 mg (absolute potassium content; Figure 1)
    minus  70% which is lost in the cooking water (Figure 2)
    minus 88% (Figure 3 bioaccessibilty raw food 12%) + 22%  (Figure 4 variation of accessibility after boiling). 12% bioaccessibilty raw + 22% cooked results in 15%)
    This would mean cooked peas deliver only about 90 mg potassium per 100 g.
    This information could be added for all food in an extra table 5 because it is of general interest. Every nutrition guideline I know of marks peas as undesirable in hemodialysis patients

A... We added Table 1 and Table 2 reporting the information requested by the reviewer.

  1. The manuscript could have been proofread by a native English speaker

A... Following the suggestion of the reviewer, the manuscript has been proofread by a native English speaker

Reviewer 2 Report

Very unique and of practical importance

my only suggestion would be to have a simple table of K bioavailabilty with the foods that are only high raw and those only high when cooked

also information on the skin of the food and the rinse water

Author Response

Reviewer no. 2

Very unique and of practical importance

my only suggestion would be to have a simple table of K bioavailabilty with the foods that are only high raw and those only high when cooked

A… Thank you for your comment. We have added, in Table 2, the bioaccessible potassium in all plant food under investigation. In this way, the reader can be defined food with high and low potassium content if raw or cooked.

also information on the skin of the food and the rinse water

A…. We have not determined potassium in different part (flesh, peel, etc.) of fruits, vegetables or legumes and we have not determined potassium leached in the rinse water.